# Functional Genomic Analysis of Breast Cancer Metastasis: Implications for Diagnosis and Therapy

**DOI:** 10.3390/cancers13133276

**Published:** 2021-06-30

**Authors:** Ziqi Yu, Mei Song, Lotfi Chouchane, Xiaojing Ma

**Affiliations:** 1Department of Microbiology and Immunology, Weill Cornell Medicine, 1300 York Avenue, New York, NY 10065, USA; mes2047@med.cornell.edu; 2Department of Genetic Medicine, Weill Cornell Medicine-Qatar, Qatar Foundation, Doha P.O. Box 24144, Qatar; loc2008@qatar-med.cornell.edu

**Keywords:** breast cancer metastasis, genomic analysis, diagnosis, prognosis, therapy

## Abstract

**Simple Summary:**

Metastasis remains the greatest cause of fatalities in breast cancer patients world-wide. The process of metastases is highly complex, and the current research efforts in this area are still rather fragmented. The revolution of genomic profiling methods to analyze samples from human and animal models dramatically improved our understanding of breast cancer metastasis. This article summarizes the recent breakthroughs in genomic analyses of breast cancer metastasis and discusses their implications for prognostic and therapeutic applications.

**Abstract:**

Breast cancer (BC) is one of the most diagnosed cancers worldwide and is the second cause of cancer related death in women. The most frequent cause of BC-related deaths, like many cancers, is metastasis. However, metastasis is a complicated and poorly understood process for which there is a shortage of accurate prognostic indicators and effective treatments. With the rapid and ever-evolving development and application of genomic sequencing technologies, many novel molecules were identified that play previously unappreciated and important roles in the various stages of metastasis. In this review, we summarize current advancements in the functional genomic analysis of BC metastasis and discuss about the potential prognostic and therapeutic implications from the recent genomic findings.

## 1. Introduction

Breast cancer (BC) is the most common tumor worldwide with around 2,260,000 new diagnoses worldwide in 2020, accounting for approximately 11.7% of all cancers [1]. It is the second leading cause of death in women with cancer and accounts for an estimated 15% of all cancer-related deaths in the US [2]. Over the last decade, pivotal technological and clinical advances dramatically improved the survival of BC patients. However, metastasis is still the most frequent cause of cancer-related deaths. Over 90% of cancer-related deaths are caused by the metastasis-related complications [3]. The 5-year overall survival rate of BC patients without metastasis is 90% [2]; however, distant metastasis can lead to a dramatic reduction of this rate to approximately 25% [4]. Metastasis is a complicated and poorly understood process that can be usually parsed into interrelated steps, beginning with cancer cells detaching from the primary tumor and entering the circulation (intravasation) [5,6]. Next, the circulating tumor cells (CTCs) that survive in the blood vessel eventually extravasate through the vascular wall of distant organs into the parenchyma, resulting in the formation of metastatic colonies [7]. Great advances in mouse models and genomic sequencing technologies brought new ways into investigating the molecular mechanisms of metastasis. Here, we summarize current findings of molecular mechanisms of BC site-specific metastasis discovered with the advent and power of genomic technologies and discuss about the potential prognostic and therapeutic implications from the current perspective.

## 2. Major Hypotheses on Metastasis

The metastatic localization is not a random process but rather at preferred sites under the control of a multitude of microenvironmental, cellular, and molecular factors [8]. The pattern of cancer spread by which primary tumors tend to metastasize to distinct organs is usually referred to as organ-specific metastasis which was first described by the “seed and soil” theory of Stephen Paget [9]. For instance, BCs have a propensity to metastasize to bone (50–65%), lung (17%), brain (16%), and liver (6%), while metastases to other organs such as spleen, kidney or uterus, are relatively rare [10], and the metastatic heterogeneity leads to varied responses to treatment and patient prognosis. The 5-year overall survival rate of bone metastasis is 22.8% [11]. Lung is the second most frequent site of BC metastasis with a 5-year overall survival of 16.8% [12]. The occurrence rate of liver metastasis is relatively low, but the survival rate is poorer relative to proximal, bone, and lung relapse, with a projected 5-year overall survival of 8.5% [13]. 

Different hypotheses of organ-specific metastasis were put forward in the past years, and the most widely accepted model of metastasis is the “seed and soil” theory. It essentially stipulates that the intrinsic properties of tumor cells and their supportive microenvironment of target tissues cooperate to determine a successful second-organ-colonization [9]. Ample clinical data showed that primary tumors of breast cancer with different histologic/molecular characteristics have the propensity to metastasize to distinct organs [12,14,15,16,17,18,19,20]. For example, while bone is the most common metastatic site in luminal A (66.6%), luminal B (71.4%), and luminal/HER2 (65%) groups and the least common site in the basal group (39%), TNBC has the greatest tendency to metastasize to the lung, accounting for about 42.8% of patients [19]. Moreover, to answer the question of when and how metastasis is initiated, a hypothesis of a presumed common ancestor was put forward and widely acknowledged. The hypothesis proposes two main models, the linear versus the parallel progression models, sometimes also referred to as a late dissemination versus an early dissemination models [4,7,21,22,23,24]. The linear progression model or late dissemination model is based on Leslie Fould’s description of tumor progression as a series of stepwise changes in several unit characteristics [25]. At its simplest, the model states that the rapid renewal of cancer cells is normally achieved by rather few divisions of a special minority population, the immortal stem cells, which pass through multiple successive mutation and selection. The rate of accumulation of mutations in such a system will be proportional to the rate at which they have to multiply to replace lost stem cells [26]. These immortal stem cell clones expand and individual cancer cells leave the primary site to seed secondary growths. Two studies of single breast cancer patients using genome sequencing provided support for the linear progression model, where metastases emerge from late occurring advanced clonal subpopulations [27,28]. A recent study comparing breast cancer metastases and primary tumors using whole-genome sequencing found a high concordance rate in single mutations between matched primary tumors and metastases. Among the synchronous lymph node metastases, only one patient had a driver mutation (in PTEN) seen in the metastasis that was not present in the primary tumor, confirming that there is generally little genomic divergence between primary and synchronous local lymphatic metastases [29]. For example, mutations of the tumor suppressor gene TP53, a common event between primary tumor and metastasis lesion, were usually early events. TP53 mutations were rare in T1 stage BCs, and significantly more frequent in T3 stage (>5 cm) tumors. Clonal expansion of TP53 mutated cells therefore often occurs when tumors grow beyond 2 cm [30]. Correlation between tumor size and metastasis provides a strong argument in favor of the linear progression model. 

The parallel progression or early dissemination model, put forward especially by C. Klein [21] and postulates that tumor cells acquire metastatic potential very early, even when the primary lesion is small or undetectable, and different tumor clones can be seeded in parallel to multiple secondary sites [31]. This model was supported for breast cancer by very recent evidence in a mouse model [32], hypothesizes that the dissemination of metastatically capable cells by the primary tumor occurs in very early stages of primary tumor development, and that the primary and a secondary lesion therefore are evolving separately thereafter. The genetic analyses of disseminated cancer cells (DCCs), healthy mammary glands, primary tumors (PT), and lung metastases of HER2 positive mouse BC model revealed that 80% of metastases are derived from early DCCs. The genomic profiles of human DCCs isolated from bone marrow of M0 stage (without manifestation of metastasis) BC patients indicated that early DCCs are yet to acquire critical alterations, such as chromosome 8q gains, which were lacking in the primary tumor to form metastases [32].

Overall, these two models depict similar but not identical perspectives that genetic and nongenetic alterations as well as selection pressures from the microenvironment give rise to heterogeneous cellular populations and metastatic potential [33]. From the research conducted to date, it is very difficult to definitely support only one of the two models for BC. BC metastasis is an evolutionary process that involves several sequential steps and diverse microenvironments, metastatic niches, and, in general, stromal components also undergo a dynamic evolution during the metastasis process [34,35,36]. Moreover, recent next generation sequencing (NGS) data indicate that breast cancer metastasis is a multidirectional process whereby cancer cells can seed distant sites, as well as the primary tumor itself [37,38]. In this context, a third mechanism, the concept of “tumor self-seeding” was proposed [38]. Kim et al. observed self-seeding in homotypic BC models and presented evidence that tumor self-seeding is a general phenomenon in breast carcinoma. Further, they found that a tumor can be seeded by CTCs from lung-metastatic nodules indicating that self-seeding is a potential cause of local recurrence after tumor excision. This model does not necessarily have to be mutually exclusive to linear or parallel progression models but can be considered to take place in parallel to contribute the distance organ metastasis and local recurrence [21,37,39] (as illustrated in Figure 1).

With the rapid and ever-evolving genomic sequencing technology-related development, a number of molecules were found to play important roles in different steps of metastasis; for example, alterations in the expression of E-cadherin and metalloproteinase 9 (MMP9), which are epithelial-mesenchymal transition (EMT) markers that play important roles in mediating tumor invasion and metastasis [40,41,42]; chemokine receptors including CXCR4 and CCR7, which play critical roles in mediating the infiltration of metastatic BC cells into the lung parenchyma and the homing of tumor cells to lung niches [43], and angiogenesis activators and inhibitors that regulate the blood supply to primary tumors and metastases [6,44,45]. Several new genes were identified that participate in TGF-β-induced EMT, including HOXB7, which can directly bind to the promoters of TGF-β2 [46]. Numerous earlier studies on BC metastasis restricted themselves to studying primary tumor tissues or often focused only on one individual gene of interest at a time; these approaches are too prone to bias to generate a comprehensive viewpoint of the molecular mechanisms regulating the metastasis cascade. As a result, the research efforts to combat or even prevent metastasis are considered rather fragmented. Latest technological developments in advancing attractive models to study metastasis, large-scale DNA microarray technology and single-cell sequencing of disseminated tumor cells (DTCs) [47,48] are revolutionizing the way we approach metastasis research.

## 3. Metastasis and Recurrence Prediction from Genomic Profiling of Primary Tumors

Traditionally, prediction of BC metastasis relies on standard descriptors and physical characteristics, such as patient age, tumor size, histological features (tumor grade), and number of involved axillary lymph nodes [49]. However, these predictors fail to make accurate predication about the course of disease for individual cancer patients, in which patient with the same stage of disease can have markedly different treatment responses and overall outcome [50]. With the improvement of early detection techniques, more and more patients with early-stage BC, particularly those with ER-positive and/or PR-positive and HER2-negative tumors, are overtreated with chemotherapy. For example, 70–80% of patients receiving chemotherapy or hormonal therapy would have survived with surgery and radiotherapy alone [51,52].

Therefore, advances in the understanding of molecular signaling pathways and genetic signatures of BC are helpful to identify more accurate prognostic markers capable of recognizing patients at risk of relapse following local therapy. Genomic profiling technology enables the identification of gene signatures to predict prognosis for BC metastasis and guide the use of adjuvant therapy. Numerous studies were carried out and described the classification and prognosis based on genomic profiling in various cancers, such as BC [53,54,55,56], prostate cancer [57], liver cancer [58], melanoma [59], leukemia, and lymphoma [60,61]. High-throughput microarray technologies capable of documenting the expression of thousands of genes simultaneously indicated that molecular diagnostics based on array profiling may have far superior performance compared with that of traditional histopathologic techniques. In 2002, van’t Veer et al. used DNA microarray analysis on primary BC tumor of 117 patients and applied supervised classification to identify a 70-gene ex-pression signature strongly predictive of a short interval to distant metastases (‘poor prognosis’ signature) in patients who were free of lymph node metastases at the time of diagnosis [54]. The 70 genes that were upregulated in the poor prognosis signature were involved in cell cycle, invasion and metastasis, angiogenesis, and signal transduction (for example, cyclin E2, MMP9 and MP1, RAB6B, and the VEGF receptor FLT1). Following the St Gallen and National Institutes of Health guidelines, up to 90% of lymph-node-negative young BC patients are candidates for adjuvant systemic treatment. The 70-gene signature would only recommend systemic adjuvant chemotherapy to only 20–30% of these patients and could avoid about 70–80% of these patients suffering unnecessary adjuvant systemic therapy. The 70-gene signature classifies tumors into groups that are associated with a good prognosis or a poor prognosis on the basis of the risk of distant recurrence at 5 and 10 years [62]. The clinical utility of the 70-gene signature was validated by several large-scale, independent studies [63,64].

In addition to the apparent value of this technique in clinical practice, a provocative implication of these studies was that all or most cells in the primary tumor have met-astatic potential, since the signature that predicts metastasis could be identified in the bulk of the tumor. BC is a heterogeneous disease; several studies of the genomic landscape of invasive BC clearly show that each BC is genomically distinct, with a high level of diversity in single nucleotide variants (SNVs), small insertions and deletions (indels), structural variants (SVs), and copy number alterations (CNAs) [65,66,67,68,69,70]. Moreover, there is still a large diversity of mutations between different tumor regions within the same individual tumor [71] (as illustrated in Figure 1). Gene expression studies examined genetic markers as prognostic factors in BC patients with brain [72], lung [73], and bone metastasis [74]. For example, Liu et al. evaluated copy number imbalances (CNIs) by whole-genome molecular inversion probe arrays and found that copy number gains at 1q41 and 1q42.12 and losses at 1p13.3, 8p22, and Xp11.3 independently increased risk of bone metastasis [74]. However, in individual cases, all lesions shared genetic alterations, suggesting that they may have a common clonal origin. This is supported by Desmedt, C and coworkers, who analyzed multiple invasive tumors from multifocal BC patients by targeted gene sequencing analysis and low coverage WGS [75]. In more than 65% of cases, all lesions shared precise genetic alterations, whilst the remaining cases shared structural/copy number variants. In 2003, van’t Veer et al. compared the gene expression between the primary breast tumors and corresponding distant metastases and found that gene expression profiles of them were strikingly similar [76]. With the development of genomic technologies, many early molecular alterations (for example, mutations of TP53, PIK3CA, CDH1, GATA3, amplification of MYC, CCND1, ERRB2/HER2) are found prevalent in metastatic deposits [77]. Even small primary tumors without lymph node metastases can display the poor prognosis signature, indicating that they are already programmed for this metastatic phenotype [54,78]. Therefore, the gene alterations favoring metastasis may exist in the bulk of malignant cells of a metastasis-prone primary tumor and are retained in its metastases.

## 4. Metastasis Prediction from Genomic Profiling of Circulating Tumor Cells

Circulating tumor cells (CTCs) are the tumor cells that are released from the formation and growth of primary tumor and/or metastatic sites into the bloodstream. Analyzing CTCs as the link between the primary tumor and metastatic sites, therefore, would give insights into the biology of the metastatic cascade. Moreover, it may serve as a valid counterpart for the assessment of prognostic and predictive factors in patients with met-astatic BC (MBC) [79]. The process of tumor cells invading the basement membrane and surrounding tissue and entering the bloodstream is a first and crucial step for metastasis. Nevertheless, the environment in the bloodstream is harsh for tumor cells owing to the physical forces, the presence of immune cells, and anoikis, and it is likely that CTCs might undergo a strong selection process which contributes only an extremely small proportion of the CTCs to forming secondary tumors [80,81,82,83]. This process is supported by the observation that only a few intact cells are found in the blood of cancer patients [84]. As the dissemination of tumor cells to distant organ sites necessitates a series of processes through the vasculature, it is fostered by close association with activated platelets and macrophages [85,86,87]. Tissue factor (TF, also known as coagulation factor III or CD142), expressed by tumor cells, triggers the formation of a platelet clot around the tumor cells. The platelet clot then triggers the localization of CD11b+ monocytes/macrophages to the tumor cells, which results in the establishment of microclots (also called CTC clusters) that protect CTCs to survive in blood [88]. These experimental studies provide a possible explanation for observational studies that show that aspirin reduces the long-term risk of BC and the risk of distant metastasis through blocking platelet aggregation [89]. Thus, platelets guard tumor cells from immune elimination and promote their arrest at the endothelium, supporting the establishment of secondary lesions [90].

The time of CTCs in the bloodstream is short (half-life: 1–2.4 h) [91]. This is one of the limitations of CTC enumeration before it extravasates into a secondary organ or is actively cleared from the blood [92]. Recent progress was made in the development of various devices to positively or negatively enrich and detect CTCs on the basis of biologic properties (i.e., expression of surface molecules) or on the basis of physical properties (i.e., size) [93,94]. The Cellsearch platform is the only FDA-approved method for the isolation and enrichment of CTCs in BC. It is a tool that can be used for positive selection of CTCs expressing cytokeratins (cytoskeletal proteins present in epithelial cells) and EpCAM (epithelial cell adhesion molecule) that are not expressed on the surrounding blood cells [95]. Cristofanilli et al. showed that CTC enumeration detected by the Cellsearch™ system defined patients into two subgroups, the group with fewer than 5 circulating tumor cells per 7.5 mL and equal to or higher than 5 circulating tumor cells per 7.5 mL [96]. Patients with ≥5 CTCs per 7.5 mL of whole blood, as compared with that of the group with <5 CTCs, had a shorter median progression-free survival (2.7 months vs. 7.0 months, *p* < 0.001) and shorter overall survival (10.1 months vs. >18 months, *p* < 0.001). The number of CTCs is an independent predictor of progression-free survival and overall survival in patients with MBC [97]. Persistence of CTCs after chemotherapy was also found to be associated with therapeutic resistance [98]. Moreover, recent data confirmed the prognostic value of CTCs in early-stage BC [99,100]. However, using CTC enumeration as part of clinical standard practice remains unapproved [101], and a human trial testing its clinical utility did not show improvement in overall survival (OS) or progression-free survival (PFS) when therapy was adjusted based on CTC enumeration [102]. These caveats highlight the importance of further CTC molecular characterization to provide more informed choices of treatment options.

The molecular characterization of CTCs could provide the information on phenotypic identification of malignant cells and genetic alteration that may change according to disease progression and therapy resistance (as illustrated in Figure 2). However, monitoring CTC characteristics was initially performed on enriched fractions [103], which provided only very limited information on tumor heterogeneity. With the advances in technologies for single-cell analysis made during the past 5 years, analyses of CTCs at single-cell resolution in peripheral blood could offer a unique minimally invasive approach to characterize and monitor dynamic changes in tumor heterogeneity in individual patients with cancer at the genomic, transcriptomic, proteomic, and functional levels [104].

CTCs can be detected at a low cutoff of 1 cell in 27% of patients at the early stage of BC [99,105], and CTC detection before and/or after neoadjuvant chemotherapy was significantly associated with early metastatic relapse. Hence, the detection and molecular characterization of CTCs in early-stage BC may contribute to the decision-making of clinicians in the selection of patients for strict follow ups for consideration of secondary adjuvant treatments [106]. Rossi and coworkers evaluated the CNA profiles of single CTCs isolated from early-stage BC patients at different time points by exploiting a whole-genome low-coverage next-generation sequencing (NGS) approach [107]. They found that CTCs persisting even months after tumor resection shared several CNAs with matched tumor tissue, suggesting the presence of regions potentially associated with their persistence. Moreover, the enrichment analyses revealed that type I interferon (IFN)-associated genes were thoroughly altered in CTCs. The IFN pathway was reported to contribute to apoptosis, cellular senescence, increased migration, and drug resistance depending on the IFN-stimulated genes transcribed in BC [108,109]. In another study, CTCs and primary tumors were profiled using RNA-seq. Strikingly, no single markers were universally present in all CTCs. But, in both cohorts, the gene signature of the shared top 75 upregulated genes between CTCs and primary tumors (PTs) was prognostic of worse overall survival [110].

Not every CTC will arrive at a secondary site, and even fewer will be able to develop into metastases. There must be different populations of CTCs, some more adept at survival than others, and some with a greater propensity to metastasize than others. How-ever, certain homogeneous genomic gains were detected at primary breast carcinomas and CTCs in MBC, highlighting occult target changes that could be responsible for the preferential passage of tumor cells into circulation [111]. Furthermore, higher number of CTCs was associated with genomic alterations in ESR1, GATA3, CDH1, and CCND1, while lower number of CTCs was associated with CDKN2A alterations in MBC [112].

Development of organ-specific metastasis is not random but rather a selective and specific process. CTCs are highly heterogeneous with some subsets capable to survive and only a few clones can interact with a specific target organ microenvironment or fostering CTC colonization at the organ site by the formation of a metastatic niche [113]. A recent study compared gene expression of sequentially generated CTC-derived xenograft (CDX)-derived cell populations, together with online gene expression arrays, and TCGA databases to discover a CTC-driven, liver metastasis-associated TNBC signature [114]. This investigation predicted 16 hub genes, 6 biomarkers with clinically available targeted drugs, and 22 survival genes. It implies that CTC molecular properties can be clinically useful tools to predict the risk of metastatic recurrence at a specific organ and to potentially drive therapy. There is very limited data on the BC organ-specific metastasis based on detection of CTCs [115,116]. Future mechanistic investigations and prospective studies are needed to delineate the role of CTCs detection in BC organ-specific metastasis.

## 5. Functional Genomic Analysis of Site-Specific Metastasis of BC

Molecular profiling of MBC typically focused on the primary breast lesion. Although sequencing of primary BC provided insight into the biology of early malignancy, the majority of the patients presenting with such a disease will not relapse after conventional therapy. Therefore, understanding the biology of early BC will not help in deciphering the specificities of the lethal disease or translate into treatment advances. Many studies revealed that metastases are clonally related to the primary tumor, sharing many of the driver mutations, but nonetheless have typically acquired additional variants not detectable in the primary lesion during progression [27,29,71,117,118,119,120,121,122,123,124]. For example, ESR1 mutations or amplification are rarely observed in primary disease but could be acquired during the disease evolution and are prominent and critical drivers of resistance to endocrine therapy [125,126,127].

A further complexity is that metastatic tumor deposits are not exact replicas of the primary tumor from which they arose in either a morphological or a molecular sense. Indeed, metastatic tumors at different sites within an individual may display widely disparate features [128,129,130]. Although the accruing sequencing data suggest that all metastases within a patient shared a common ancestor, significant intermetastasis heterogeneity is invariably observed within patients [120,129,131,132,133]. De Mattos–Arruda et al. used whole-exome sequencing (WES) to characterize the somatic mutational landscape across 79 metastases and 7 body fluid samples, with a range of 2 to 19 metastatic samples per patient sequenced. They found that driver and nondriver gene mutations were heterogeneously accumulated in different metastases [131]. Here, we describe the advances in understanding of molecular alterations of different tropisms in BC metastasis, including, bone, lung, liver, and brain, respectively (as illustrated in Table 1).

### 5.1. Bone Metastasis of BC

Bone metastasis develops in approximately 70% of patients with advanced BC and contributes to significant morbidity due to pain and skeletal related events (SREs) [140]. Remodeling occurs constantly in the healthy skeleton to regulate calcium homeostasis to repair damage to the bone and withstand new external stresses to the skeleton. The receptor activator of nuclear factor-kB ligand (RANKL) is a major regulator of bone mass. Studies showed that RANK is expressed on the surface of BC cells and directs BC cells to metastasize to bone [141]. When BC cells metastasize to the bone, they and their circulating factors may affect bone stromal cell targeting, which disrupts the normal bone homeostasis and starts a vicious cycle.

The complex reciprocal effects between tumor cells and the bone stomal cells contribute to tumor cells metastatic homing and outgrowth. To identify relevant factors, a study of microarray gene-expression analysis of a cohort of metastasis samples that were surgically removed from BC patients, including 16 metastases from bone, 18 from lung, 19 from brain, and 5 from liver [134] identified 17 genes that were expressed in bone metastases at a higher level (>2-fold) than in metastases from other sites. These genes include the C-X-C motif chemokine 12 (CXCL12)/stromal cell-derived factor 1 (SDF1), transforming growth factors β (TGFβ), insulin-like growth factors (IGFs), jagged 1 (JAG1), vascular endothelial growth factor C (VEGFC), etc. CXCL12 is predominantly produced by various bone stromal cells, and acts as the guardian for tumor cells expressing the receptor, CXCR4 [43]. JAG1 is an important mediator of bone metastasis by activating the Notch pathway in bone cells. Notch-ligand jagged in tumor cells promotes survival and proliferation by stimulating IL-6 release from osteoblasts or promoting bone metastasis cytokine TGF-β release during bone destruction [142].

### 5.2. Lung Metastasis of BC

Only a small percentage of tumor cells that disseminate via circulation can survive at distant sites and form micrometastases [143]. Most intravenously injected cancer cells that lodge in the lung will die within two days [144], which may be mainly attributed to immune attacks by leukocytes [145]. The first barrier before tumor cells entering lung microenvironment is the tight cell-cell junctions caused by continuous endothelial cells. Metadherin (MTDH), a cell surface molecule with an extracellular lung-homing domain (LHD), could mediate lung-homing of BC cells through binding to a receptor expressed by the lung endothelium and enable efficient transmigration of cancer cells into the lung parenchyma [146]. After extravasation into the lung parenchyma, metastatic tumor cells which are qualified with the ability of survival and adaptation to a new microenvironment can form the new niche. To search for mediators of lung metastasis, Minn et al. performed a transcriptomic microarray analysis and identified an 18-gene lung metastasis signature (LMS) expressed in BC cells [73]. To identify which of the genes in the LMS signature are able to confer growth advantages exclusively in the lung microenvironment, they knocked down various lung metastasis genes that were previously assayed for effects on metastatic behavior. IL13Rα2, SPARC and VCAM1 were found to decrease lung metastatic ability but not orthotopic tumor growth. In xenograft model systems, VCAM1 tethers macrophages to cancer cells via counter-receptor α4-integrins to evade apoptosis in a leukocyte-rich microenvironment, and this interaction triggers the activation of Ezrin, which subsequently activates PI3K-AKT signaling in cancer cells, thereby increasing their survival [147].

After evasion from apoptosis, the mutual regulation between tumor cells and other cells in the lung parenchyma facilitate their proliferation and the formation of de novo niches that support metastatic outgrowth. Tumor cells could educate lung fibroblasts to produce the extracellular matrix (ECM) component periostin (POSTN) [148] and tenascin C [149], which subsequently activate WNT and Notch signaling, respectively, which may help them to maintain their stemness properties and enhance metastatic colonization. Transitions between epithelial and mesenchymal states have crucial roles in cancer. The disseminated mesenchymal tumor cells recruited to the target organs may undergo mesenchymal to epithelial transition (MET), which would favor metastases formation [150]. Gene expression profiling revealed that the myeloid cells from metastatic lungs express versican, an extracellular matrix proteoglycan. Versican stimulated MET of metastatic tumor cells by attenuating phospho-Smad2 levels, which resulted in elevated cell proliferation and accelerated metastases [135]. Moreover, Huber et al. found that NF-κB activity was necessary for cells to be maintained in a mesenchymal state, as its inhibition causes reversal of EMT, leading to the formation of lung metastases by H-Ras-transformed epithelial cells [136].

### 5.3. Liver Metastasis of BC

In a recent work that analyzed the targeted sequencing results of the Memorial Sloan–Kettering Cancer Center (MSKCC) dataset of liver metastasis and primary BC extracted from the MSK-IMPACT Clinical Sequencing Cohort, mutations specific to the metastasis were enriched within the PI3K-AKT pathway molecules [137]. The importance of the activation of the PI3K-AKT pathway in hepatic metastases of colorectal cancer was already demonstrated [151]. Moreover, Toy et al. observed that TP53, PIK3CA, and GATA3 also had the most frequent mutations in other metastasis organs of BC, including brain, liver, lung, and bone [152]. For example, Zhang et al. reported that in bone metastasis of BC, CXCL12/SDF1 and IGF1 positively selected tumor cell clones with elevated proto-oncogene tyrosine-protein kinase sarcoma (SRC)-activity, which in turn led to PI3K-AKT pathway activation and increased survival [134]. The mechanisms of PI3K-AKT pathway during the liver metastasis in BC remain to be fully delineated.

The motility and invasiveness of cancer cells that undergo EMT may favor their dispersion to distant organs. Oskarsson et al. surmised that cancer cells that underwent an EMT for metastatic dissemination must traverse the reverse process, a mesenchymal-epithelial transition (MET), to initiate metastatic colonization [153]. Recent research found that BC liver metastases displayed unique transcriptional fingerprints, characterized by down-regulation of extracellular matrix (i.e., stromal) genes [138]. Other studies also showed that N-cadherin, fibroblast growth factor receptor (FGFR) and MMP-9 boosted proliferation and activated liver metastasis to overcome the suppressive effect of E-cadherin [154]. Similarly, the homeobox factor Prrx1 is an EMT inducer confer-ring migratory and invasive properties of BC cells, whereas the subsequent downregulation of Prrx-1 induced an MET that facilitated metastatic colonization without suppressing stem cell traits [155]. This observation adds to the picture that EMT/MET-orchestrating processes are essential for metastasis, specifically metastasis of BC to the liver.

### 5.4. Brain Metastasis of BC

Metastatic cells invading the CNS parenchyma must pass the blood-brain barrier (BBB), a semipermeable barrier to metastasis, which is comprised of endothelial cells, astrocytes, and pericytes forming the neurovascular unit [156]. One of the prominent obstacles for cancer cells to colonize the brain is extravasation across the BBB. Bos et al. per-formed a genome-wide expression analysis of brain metastatic cells and parental cells in a BC brain metastases (BCBM) mouse model, which identified 17 genes that were specifically correlated with brain relapse. Among them, cyclooxygenase-2 (COX2), α-2,6-sialyltransferase (ST6GALNAC5), and the epidermal growth factor receptor (EGFR) ligand HBEGF are major mediators of brain metastasis [72]. COX2 upregulates the expression of MMP-1 in brain metastasis of BC patients to promote angiogenesis [157]. Moreover, COX2 expression is associated with BBB permeability. COX-2 expression promotes BC metastases via upregulating miR-655 and miR-526b and the induction of stem-like cells (SLC) through EP4-mediated signaling [158,159]. ST6GALNAC5 over-expression is able to promote transmigration of BC cells through HUVEC endothelial cells, a brain-like endothelial barrier that mimic the BBB in vitro model. ST6GALNAC5 expression in human cancer cells (MDA-MB-231) resulted in the accumulation of GD1α at the cell surface, leading to a lower degree of adhesion between BC cells in a human BBB model [160].

Once infiltrated into the brain parenchyma, cancer cells encounter a number of host cell types, including pericytes, reactive glia, neural progenitor cells, neurons, and oligodendrocytes [161]. Astrocytes, the predominant glial cells in the central nervous system (CNS), are responsible for homeostasis of the brain microenvironment. Once normal astrocytes encounter cancer cells, they become reactive astrocytes (RAs) and limit the survival of arriving metastatic cells at the initial stages of BCBM [162]. As a result, most cancer cells are unable to tolerate the neuroinflammatory reaction that is rapidly instigated by microglia and astrocytes [163,164]. However, recent evidence demonstrates astrocytes can protect BC cells from chemotherapeutic agents through upregulation of survival genes [165]. In BC patients, large numbers of glial cells were found within the inner tumor mass of metastatic foci [166]. The colonization and formation of BCBM depends on interactions between the microenvironment and the colonizing metastatic BC cells [167]. Molecular profiling of paired brain metastases and corresponding primary breast tumors by whole-exome sequencing revealed that brain metastases harbored part of genomic aberrations in the PI3K/AKT/mTOR pathways, which were not detected in the corresponding primary tumor [121]. Tumor cells can remarkably optimize the brain microenvironment by inducing growth factor receptors and activating AKT/PI3K/mTOR signal pathways [168]. In a recent study using NanoString nCounter Analysis covering 252 target genes to compare gene expression levels between primary BCs that relapsed to brain and BCBM samples [139]. SOX2 and OLIG2 mRNA expression was increased in BCBM compared with that of the primary BC. SOX2 and OLIG2 are the key transcriptional factors expressed in developing and mature central nervous system (CNS) and control the reprogramming human stem cells [169,170]. Park et al. showed that the interactions with astrocytes might contribute to the reprogramming of the cancer cell transcriptome, resulting in a gain of neuronal cell characteristics [171]. Moreover, RNA-Seq was performed, yielding a fundamental circRNA profile of BCBM, which serves as a framework for better understanding of the mechanisms inducing brain metastasis and for identifying future biomarkers and therapeutic targets of clinical interest [172].

## 6. Therapeutic Implications of the Genomic Information

In the past decade, significant efforts were made to characterize the genetic drivers in BC metastasis using modern sequencing techniques. Better understanding of the genomic complexity and heterogeneity of BC metastasis will lead to improved treatment strategies and research directions. Moreover, the extent of response and potential resistance to treatments varies among different metastatic types and patients. Therefore, more effective and targeted treatments that build on the molecular mechanism of meta-static BC are needed.

Given that MTDH mediates the adhesion of cancer cells to lung endothelium, many ways against MTDH were developed to inhibit lung metastasis of BC, e.g., the antibodies reacting to the LHD of MTDH, the multiple tyrosine kinase inhibitor (TKI) SU6668, and DNA vaccines. Denosumab, a mAb directed against RANKL, reduces the SRE risk through inhibiting bone destruction caused by RANKL [173,174]. Lapatinib is a TKI that acts against both HER2 and EGFR receptors, which are major mediators of BM [71], with a demonstrated ability to cross the BBB. In a multicenter phase II study, the addition of capecitabine to lapatinib resulted in an encouraging intracranial response rate and survival [175]. In addition to molecules that mainly mediate single-organ metastasis, several molecules exhibit pleiotropic functions in metastasis to multiple organs. Actionable mutations in PI3K/AKT/mTOR pathway are frequent in bone, lung, live and brain metastasis of BC patients from above sequencing evidence. The addition of everolimus, an mTOR inhibitor, to an aromatase inhibitor in patients with hormone receptor positive metastatic BC and to trastuzumab and vinorelbine in patients with HER2-positive BC led to im-proved survival outcomes in randomized placebo-controlled phase III trials [176,177]. Overexpression of CCL2 promotes BC metastasis to both lung and bone in a manner de-pendent on its receptor CCR2 expressed on stromal cells to increase macrophage infiltration and osteoclast differentiation. Blocking CCL2 function with a neutralizing antibody can reduce lung and bone metastases [178].

Moreover, metastasis is a multigenic process, establishing the functional role of candidate metastasis genes may not be easily accomplished by introducing individual genes into weakly metastatic cells to enhance their metastatic phenotype. With the development of whole genome sequencing, more metastasis-related microRNAs (MiRNAs) were identified. MiRNAs, 20–22 short nucleotide sequences that often negatively regulate gene expression, can regulate multiple genes and hence multiple processes simultaneously. Since miRNA can target multiple sets of genes, it is an excellent clinical choice for cancer metastasis, a process mediated by multiple deregulated genes. For example, miR-7 was shown to prevent BC cell spreading but also to inhibit tumor-associated angiogenesis in the metastatic BC by downregulating EGFR [179]. MiR-7 also inhibits the abilities of invasion and self-renewal of BC stem-like cells BCBM by modulating KLF4 ex-pression [180]. Various drugs targeting tumor microenvironment components were already investigated in clinical trials, including antiangiogenic therapies, anti-inflammatory therapies, immunotherapies, and combination therapy [181].

## 7. Future Perspectives

Identification of genomic alterations specific to different metastases and targeted therapies against these mutations represent an essential research area to potentially im-prove survival outcomes for BC patients who develop metastases. One apparent disadvantage of DNA array- and RNA-seq based gene expression profiling is the limitation at the transcription level. Since protein expression and function are influenced by post-transcriptional regulations, DNA/RNA sequencing cannot provide a full view of the process of metastases. Current proteomic technologies were used to identify and characterize the molecular components of the invadopodia linked to the metastatic progression of BC cells [182]. High throughput protein arrays have also been used to compare proteins expression between normal and malignant breast tissue with some interesting findings [183]. Therefore, combining genomic profiling with proteomic analysis will further enhance our understanding of BC metastasis mechanisms, which will aid in the development of more efficacious therapeutic strategies targeting BC metastasis.

## 8. Conclusions

The systematic discovery of specific changes contributing to metastases at the genome level enables a deeper understanding of metastasis evolution and new ways for metastasis prognosis and therapy.

## Figures and Tables

**Figure 1 cancers-13-03276-f001:**
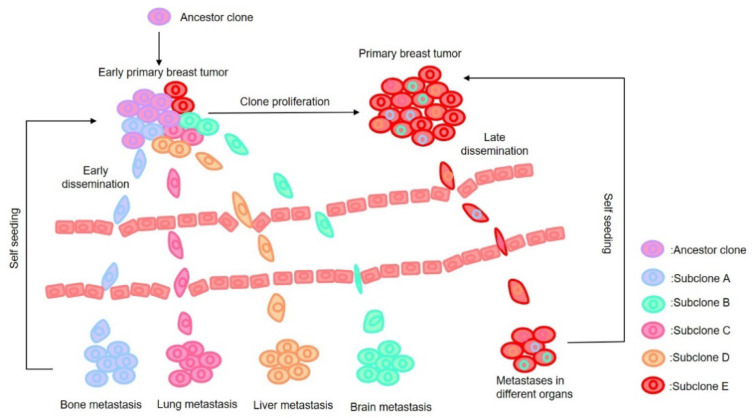
Models of metastasis evolution for BC. Based on current data, we propose a common ancestor clone and late dissemination versus an early dissemination model. Only a small fraction of tumor cells in primary site are capable of metastasizing (subclone A\B\C\D). Tumor cells can acquire oncogenic events that determine its metastatic tendency and shed from primary tumor very early. BC cells within an individual patient is heterogeneous (early primary breast tumor and primary breast tumor) and tumor cells with poor-prognosis signature (subclone A\B\C\D) are more likely to metastasize than tumor cells with good-prognosis signature (ancestor clone\Subclone E). Tumor cells derived from metastatic organ can reseed into local site.

**Figure 2 cancers-13-03276-f002:**
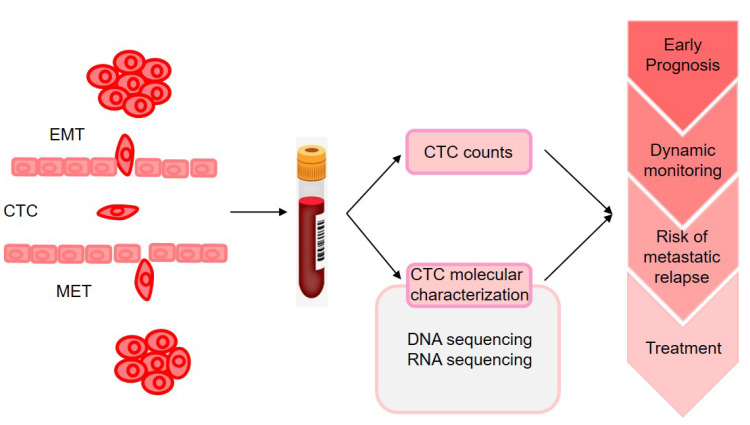
Clinical applications of CTCs sequencing for personalized medicine. Blood samples can be analyzed repeatedly for early prognosis, metastatic progression, monitoring the efficacy of therapies, and resistance mechanisms. CTC molecular characterization predict risk of metastatic recurrence at a specific organ and drive therapy potentially. EMT: epithelial-mesenchymal transition; MET: mesenchymal to epithelial transition.

**Table 1 cancers-13-03276-t001:** Molecular alteration in bone, lung, liver, and brain metastasis of BC.

The Site of Metastasis	Study	Genes	Expression Status
Bone	Latent bone metastasis in breast cancer tied to Src-dependent survival signals [134]	CXCL12/SDF1; BMP2; IGF2; CXCL14; GMFG; IGF1; JAG1; NOV; PDGFA; PGF; VEGFC; TNFSF10; TGFB1; TGFB3; SPP1; PXDN; CLEC11A;	Upregulated
Lung	Genes that mediate breast cancer metastasis to lung [73]	SPARC; IL13RA2; VCAM1; MMP2; MMP1; CXCL1; ID1; COX2; EREG	Upregulated
	Myeloid progenitor cells in the premetastatic lung promote metastases by inducing mesenchymal to epithelial transition [135]	Versican	Upregulated
	NF-κB is essential for epithelial-mesenchymal transition and metastasis in a model of breast cancer progression [136]	NF-κB	Downregulated
Liver	Prognosis and Genomic Landscape of Liver Metastasis in Patients With Breast Cancer [137]	ESR1; AKT1; ERBB2; FGFR4	Upregulated
	Transcriptional Profiling of Breast Cancer Metastases Identifies Liver Metastasis-Selective Genes Associated with Adverse Outcome in Luminal A Primary Breast Cancer [138]	MFAP5; CDH11; MMP13; FBN1; MXRA5; SFRP4; COL1A2; DPYSL3; EMP1; COL11A1; SPON1; FNDC1; RUNX2; COL3A1	Downregulated
Brain	Genes that mediate breast cancer metastasis to the brain [72]	ANGPTL4; PLOD2; COL13A1; COX2; PELI1; MMP1; B4GALT6; HBEGF; CSF3; RGC32; LTBP1; FSCN1; LAMA4; ST6GALNAC5	Upregulated
		TNFSF10; RARRES3; SCNN1A; SEPP1	Downregulated
	Genomic Characterization of Brain Metastases Reveals Branched Evolution and Potential Therapeutic Targets [122]	CCNE1; EGFR; MYC; EZH2; PIK3CA	Upregulated
	Gene Expression Profiling of Breast Cancer Brain Metastasis [139]	SOX2; OLIG2	Upregulated
		CXCL12; MMP2; MMP11; VCAM1; MME	Downregulated

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
