# Peer review of "Functional Genomic Analysis of Breast Cancer Metastasis: Implications for Diagnosis and Therapy"

_cancers, 2021, doi:10.3390/cancers13133276_

Round 1

Reviewer 1 Report

The manuscript entitled “Functional genomic analysis of breast cancer metastasis: Implications for diagnosis and therapy” reviewed models of metastasis followed by the summary of gene signatures in primary tumor and circulating cancer cells to predict the prognosis of breast cancer (BC). Given the nature that metastasis is a complicated cascade, focusing on individual genes of interest is not sufficient to generate comprehensive view of the molecular mechanism. The authors thus reviewed recent approaches and findings in BC metastasis research using advanced animal models, high throughput technologies such as DNA microarray and single-cell sequencing. The genomic analysis of site-specific metastasis of BC was also discussed. Finally, the therapeutics implications based on the genomic information were provided. The paper thoroughly reviewed most of the current advancements and understanding on BC with respect to metastasis. In general, this is an interesting and informative review article and their contributions to the field are welcome. It will indeed be very interesting to understand BC metastasis in functional genomic analysis point of view, and understand the underlying molecular mechanism of genes that are involved. However, there are some minor concerns that should be addressed before published on Cancers.

  1. In the second part of the article, various models of metastasis were presented. However, there is lack of discussion between these models (especially the 3rd “tumor self-seeding” model) and how they can be used together to explain the current metastatic status of BC. A graph contains the brief explanation of these 3 models might improve the understanding.
  2. The gene signatures of metastasis to specific target organs were well described in the article. The authors might consider provide a summary table containing the genes or biomarkers for individual organ-specific metastasis, which would further improve the comprehension.
  3. There are different types of BC, such as luminal A, luminal B, luminal HER2, and basal. Usually, luminal type adenocarcinomas are less aggressive and have better prognosis, while basal type triple negative breast cancer (TNB) are more aggressive and have worse prognosis. In this case, would TNB cells show the gene signature of “poor prognosis signature”? Could the authors provide evidences and discuss the association of genes expressed in various metastasized distal organs in respect to patients with different types of BC such as luminal, basal…etc.?
  4. Although the therapeutic implications of the genomic information was discussed in the end of the article, however, more insights and implications of the therapies and diagnosis methods need to be elaborated further to make the content more comprehensive.

Overall, the article is very well written and provides ample evidences. Additional summary table or diagram would make the article easier to read. Further elaboration on the implications and insights from the genomic information will be appreciated.

Author Response

Thank you for your time and effort that you have put into assessing the previous version of the manuscript. Please see the attachment.

Reviewer 2 Report

In this review, the authors focus on the metastatic process in breast cancer, which is responsible for most BC-related deaths. The mechanisms involved in the formation of metastasis from cells derived from the primary tumour sites in the breast are still poorly understood. However, with the development and application of genomic sequencing technologies, many novel molecules  have been identified that play pimportant roles in  various stages  of metastasis. Here, the authors summarize current advancements in the functional genomic analysis of BC metastasis and discuss about the potential prognostic and therapeutic implications.It is particularly useful the detailed description of the molecules related with each metastasis site and the information that they give on the circulating breast cancer cells prior to the seeding to form the metastasis. I was left wondering why other molecules were left out of this review, such as the products of the HOX clusters. Interesting articles spoke about circulating HOTAIR (for example)  doi: 10.1111/1759-7714.12373; or particular HOX codes in the breast cancer circulating cells doi: 10.3390/cancers13010010; 10.2174/1566524016666160316145715. Thus, in summary, I would find very interesting the addition of a part with the molecular information available also for the circulating cells, refering for example to the HOX products among others. I also found important to consider and cite this very recent report https://doi.org/10.1038/s41416-021-01327-8. Apart from that, the article is extremely nice and very useful.

Author Response

Thank you for your effort that you have put into assessing the previous version of the manuscript and your recommended articles. Both of these are great. I have carefully study the papers and cited doi:10.3390/cancers13010010 in the section 2 paragraph 5 and doi:10.1038/s41416-021-01327-8 in the section 4 paragraph 4.